# Thiamine Demonstrates Bio-Preservative and Anti-Microbial Effects in Minced Beef Meat Storage and Lipopolysaccharide (LPS)-Stimulated RAW 264.7 Macrophages

**DOI:** 10.3390/ani12131646

**Published:** 2022-06-27

**Authors:** Anis Ben Hsouna, Alex Boye, Bouthaina Ben Ackacha, Wissal Dhifi, Rania Ben Saad, Faiçal Brini, Wissem Mnif, Miroslava Kačániová

**Affiliations:** 1Laboratory of Biotechnology and Plant Improvement, Centre of Biotechnology of Sfax, B.P 1177, Sfax 3038, Tunisia; akachabouthaina@gmail.com (B.B.A.); raniabensaad@yahoo.fr (R.B.S.); faical.brini@cbs.rnrt.tn (F.B.); 2Department of Life Sciences, Faculty of Sciences of Gafsa, Zarroug, Gafsa 2112, Tunisia; 3Departments of Medical Laboratory Science, School of Allied Health Sciences, University of Cape Coast, Cape Coast 03321, Ghana; boye_alex@yahoo.com; 4Laboratory of Biotechnology and Valorisation of Bio-Geo Ressources, Higher Institute of Biotechnology of Sidi Thabet, BiotechPole of Sidi Thabet, University of Manouba, Ariana 2020, Tunisia; wissal_d2002@yahoo.fr; 5Department of Chemistry, Faculty of Sciences and Arts in Balgarn, University of Bisha, P.O. Box 199, Bisha 61922, Saudi Arabia; 6Higher Institute of Biotechnology of Sidi Thabet (ISBST), BVBGR-LR11ES31, Biotechpole Sidi Thabet, University of Manouba, Ariana 2020, Tunisia; 7Institute of Horticulture, Faculty of Horticulture, Slovak University of Agriculture, Tr. A. Hlinku 2, 949 76 Nitra, Slovakia; 8Department of Bioenergy, Food Technology and Microbiology, Institute of Food Technology and Nutrition, University of Rzeszow, 4 Zelwerowicza St, 35601 Rzeszow, Poland

**Keywords:** thiamine, anti-inflammatory, antimicrobial activity, *Listeria monocytogenes*, shelf-life extension

## Abstract

**Simple Summary:**

Thiamine (TA), also known as vitamin B1, is an essential amino acid derived from food sources for normal body function. TA is thought to have antioxidant, antimicrobial, and anti-inflammatory effects in addition to its nutritional benefits. The degree to which a number of microorganisms implicated in food rotting are sensitive to increasing concentrations of thiamine (TA), was examined. TA at increasing concentration was incubated with minced beef and then physicochemical and microbiological assessments were conducted. LPS-stimulated RAW264.7 cells were used to test TA’s anti-inflammatory capabilities. Western blot analysis revealed the expression of cyclooxygenase-2 (COX-2) and inducible nitric oxide synthase (iNOS). Finally, the ability of TA to act as a natural preservative was evaluated.

**Abstract:**

This study assessed the anti-inflammatory effect of thiamine (TA) in lipopolysaccharide-stimulated RAW264.7 cells and also assessed the preservative properties of TA in minced beef. TA demonstrated a concentration-dependent antimicrobial effect on microbial contaminants. Inhibition zones and MIC from the effect of TA on the tested bacterial strains were respectively within the ranges 15–20 mm and 62.5–700 µg/mL. TA significantly (*p* < 0.05) decreased all the pro-inflammatory factors [(nitric oxide (NO), prostaglandin E2 (PGE2), TNF-α, IL-6, IL-1β, and nuclear factor-κB (NF-κB)] monitored relative to LPS-stimulated RAW264.7 cells. TA inhibited the expression of both iNOS and COX-2. In minced beef flesh, the growth of *Listeria monocytogenes* was inhibited by TA. TA improved physicochemical and microbiological parameters of stored minced beef meat compared to control. Principal component analyses and heat maps elucidate the quality of the tested meats.

## 1. Introduction

Food contamination by pathogenic microorganisms is a major cause of food-borne illnesses and also contributes to global decline in food quality. To ensure food safety and also to improve food wholesomeness, there is the need to consider alternative food preservation methods, particularly those derived from natural sources [1]. However, several problems have been reported and are due to inefficiency of treatment processes and great resistance established by some bacteria, with pathogenic *Escherichia coli* O157:H7, *Listeria monocytogenes*, and several *Salmonella* serovars as an example [2,3]. Many food-borne diseases have been on the increase [4,5]. Meat is more susceptible to activities of bacteria involved in food spoilage. Spoilage bacteria, in particular, contribute to the cost of keeping perishable foods fresh by causing discoloration, gas production, slime production, off-odors, and off-flavors [6]. Inflammation is an overt reaction to different stimuli [7]. Despite its preventive role, too much of it can lead to cancer, atherosclerosis, and diabetes, among other chronic inflammatory diseases [8,9]. Activated macrophages are the main producers of the mediators of inflammation [10].

Cellular responses acting as powerful immunity stimulators are generated by a significant component of the cell wall of Gram-negative bacteria: lipopolysaccharide (LPS) [11]. Use of LPS-stimulated macrophages has become one of the accepted models for studying anti-inflammatory agents. Inflammatory reactions are mediated by nitric oxide (NO) and prostaglandin E2 (PGE2), which are produced by inducible nitric oxide synthase (iNOS) and cyclooxygenase- (COX-)2, respectively [12]. NO, which is produced by iNOS, is a key messenger in a variety of physiological processes, including neuronal transmission, vasodilation, and host defense. It contains antiparasitic, antiviral, micro-biocidal, and antitumoral effects [13]. However, the progression of many inflammatory diseases has been reported to be related to an excessive and sustainable release of NO, which may cause damage for the host. Additionally, inflammatory cytokines are largely a tissue response to injury. Therefore, inflammation control mediators may help to treat numerous inflammatory disorders.

Thiamine (TA), often known as vitamin B1, is a necessary nutrient for sustaining proper bodily function. It can be found in a number of thiamine-rich foods. They must be obtained from the diet because mammalian cells are unable to make them. The neurological system, heart health, energy production, and psychological wellness all benefit from TA [14,15,16]. It works as an antioxidant, scavenging free radicals and lowering the level of lipid peroxidation [17,18].

From exhaustive literature search, no reports are available on the bio-preservative and antimicrobial properties of thiamine (TA). Our research looked into the antimicrobial effects of TA on a variety of food-borne pathogens, including Gram-positive and Gram-negative bacteria and *L. monocytogenes*, as well as the bio-preservative capability of TA on raw minced beef meat. In addition, LPS-stimulated RAW 264.7 murine macrophages were used to investigate TA’s anti-inflammatory capabilities in vitro. Also, LPS-stimulated RAW 264.7 murine macrophages were used to investigate TA’s anti-inflammatory capabilities in vitro. Surprisingly, the findings of these experiments confirm TA’s potential as a natural bio-preservative and antibacterial agent with translational promise.

## 2. Materials and Methods

### 2.1. Chemicals and Reagents

Mueller Hinton broth (MBH, Bio-Rad, Marnes-la-Coquette, France), di-methylsulfoxide, sterile water, gentamicin, Thiamine (Sigma, St. Louis, MO, USA), Thiazolyl Blue Tetrazolium Bromide (MTT), DMEM, LPS, FBS, PBS, streptomycin, penicillin were purchased from Sigma–Aldrich.

### 2.2. Acquisition of Test Microorganisms and Cell Culture

Thiamine’s antibacterial action was tested on Gram-positive (G^+^) and Gram-negative (G^−^) bacteria. American type cell culture (ATCC) and the culture collection of the Centre of Biotechnology in Sfax, Tunisia, were used to obtain authentic pure bacterial cultures; *Bacillus subtilis* (ATCC 6633), *Bacillus cereus* (ATCC 14579), *Staphylococcus aureus* (ATCC 25923 and 6536), *Staphylococcus epidermidis* (ATCC 12228), *Enterococcus faecalis* (ATCC 29212), *Micrococcus luteus* (ATCC 1880), *Listeria monocytogenes* (food isolate 2132), *Escherichia coli* (ATCC 25922 and 8739), *Pseudomonas aeruginosa* (ATCC 9027), *Salmonella enterica* (food isolate), and *Klebsiella pneumoniae* (ATCC 10031). The bacterial progenies were cultured on MHB (Bio-Rad, France) at a temperature of 37 °C for 12–14 h. Inoculum preparation was done from an overnight broth culture by dilution in saline solution to 10^6^ colony-forming units CFU/mL [19].

### 2.3. Agar Diffusion Method

Antibacterial tests were carried out using the agar well diffusion method [20] and a broth micro-dilution assay for bacterial strains using sterile MH media (Bio-Rad, France). TA was dissolved in a 1:1 mixture of indi-methylsulfoxide and water and then diluted to a final concentration of 10 mg/mL. Each well plate was injected with bacterial strains and incubated at 37 °C for 24 h. Gentamicin (10 µg/well) and DMSO were employed as controls, respectively. In triplicates, inhibition zones were assessed for each group.

### 2.4. Determination of MIC and MBC Determination

The minimum inhibitory concentrations (MICs) of 13 test microorganisms from distinct species and environments were determined as previously described [19,21]. We utilized 100 µL/well sterile 96-well microplates. Test microorganisms grown in media were used as positive growth controls in wells. As a negative control, dimethylsulfoxide/water (1/9) was utilized. For bacterial strains, plates were covered with sterile plate covers and incubated at 37 °C for 24 h. The minimum inhibitory concentration (MIC) is the lowest concentration of TA at which the test microorganism does not show detectable growth during incubation. As an indicator of test microorganism growth, 25 µL of thiazolyl blue tetrazolium bromide (MTT) in indicator solution (0.5 mg/mL) dissolved in sterile water were added to the wells and incubated at 37 °C for 30 min. The colorless tetrazolium salt was reduced to a red-colored formazan product by biologically active organisms, whereas the solution remained clear after incubation with MTT when microbial growth was inhibited. As a negative control, DMSO and water were utilized. The lowest concentration of TA killing test microorganisms was established as the Minimum Bactericidal Concentration (MBC). By sub-culturing on blood agar plates without TA, this was determined from broth dilution minimum inhibitory concentration (MIC) testing. These procedures were repeated three times in each case for MIC and MBC determination.

### 2.5. Cell Viability Assay

An MTT test was used to determine cell viability [19,22]. The cells were seeded in a 96-well plate, treated with different doses of TA, and then stimulated for 18 h with 0.5 µg/mL LPS. At 37 °C in a humidified atmosphere of 5% CO_2_, cell culture was carried out in DMEM media supplemented with 10% heat-inactivated FBS and 1% streptomycin/penicillin.

### 2.6. Measurement of NO, PGE_2_, TNF-α, IL-1β, and IL-6 from Cell Extracts

The Griess reaction was used to determine NO production [19,22]. RAW264.7 cells were cultured in LPS (0.5 µg/mL) with and without TA (10, 20, and 50 µg/L) for 18 h. The amount of PGE2 produced, TNF-, IL-1, and IL-6 were measured according to methods described by yang et al., 2012 [23].

### 2.7. Western Blot Analysis

RAW264.7 cells were washed three times in 50 µL of PBS before being lysed with lysis buffer. Separated proteins were placed on nitrocellulose membranes. Membranes were treated with a secondary antibody coupled to horseradish peroxidase after an overnight incubation with a primary antibody. The immuno-reactive proteins were identified using an enhanced chemi-luminescence system (GE Healthcare, Little Chalfont, Buckinghamshire, UK) after the membranes were washed three times [19].

### 2.8. Assessment of Thiamine on Preservation of Meat

#### 2.8.1. Preparation of Raw Meat Samples

A local supplier delivered raw flesh beef (slaughterhouse of Sfax, Tunisia). To maintain its integrity, the sample was maintained in insulated polystyrene boxes on ice for an hour before being transported to the laboratory. The fresh beef was minced using a sterile grinding machine. Following that, the minced meat beef was divided into four groups: a negative control (without TA), TA (0.1 and 0.5 percent) groups, and a positive control (BHT, a synthetic antioxidant). Each treatment was held at 4 degrees Celsius for 14 days in a petri plate. On days 0, 3, 7, and 14, the quality of the product was evaluated. To avoid microbiological contamination, the meat samples were prepared according to strict sterilizing techniques. On the basis of a prior study [24,25], the TA content and beef preservation time were calculated.

#### 2.8.2. Physicochemical Analysis

##### pH Determination

5 g of minced meat sample was homogenized in 50 mL of distilled water (pH 7.00) and filtered at each sampling stage. The pH of the filtrate was determined using a pH meter, as described before [26].

##### Thiobarbituric Acid Reactive Substances Value (TBARS)

In minced beef samples, lipid oxidation was assessed by detecting TBARS using the distillation method, as previously described [27]. The results are given in milligrams of malonaldehyde equivalents per kilogram of sample (mg/kg).

### 2.9. Assessment of Microbiological Parameters

A 25 g sample of each minced beef was diluted in 225 mL sterile peptone water (Peptone 10 g/L, NaCl 5 g/L, disodium phosphate (Na_2_HPO_4_) 3.5 g/L, and monopotassium phosphate (H_2_KO_4_P) 3.5 g/L) and homogenized for 90 s at room temperature in a stomacher. In 0.1 g/100 g peptone water, a 10-fold serial dilution series was created. Total viable count (TVC; plate count agar incubated at 30 °C for 48 h), psychrotrophic bacterial count (PTC; plate count agar incubated at 7 °C for 10 days), and Enterobacteriaceae (Violet Red Bile Glucose (VRBG) agar; incubated at 37 °C for 48 h) were all used to determine the microbiological quality of each sample. To see how *L. monocytogenes* (ATCC 19117) reacted to TA over a 14-day period at 4 °C, minced meat samples were infected with the bacteria. *L. monocytogenes* cell suspension (100 mL) containing 10^6^ CFU/mL was injected into minced meat samples and properly mixed. After 0, 1, 3, 6, 8, 10, 12, and 14 days, the stored samples were examined. After 24 h of incubation at 30 °C, the colonies were counted and the *Listeria* strains were enumerated on PLACAM agar (Oxoid). Only plates holding between 15 and 150 colonies were chosen. A total of 30–300 colonies were enumerated on plates. We transformed microbiological data into logarithms of colony-forming units per gram (CFU/g).

### 2.10. Sensory Evaluation

Eighteen professional panellists were chosen from the University of Sfax’s employees to independently evaluate the minced beef samples’ color, look, odor, and overall acceptability. The evaluation was conducted on a 9-point hedonic scale, with 9 meaning “like extremely”, 1 meaning “highly dislike”, and 5 (neither like nor dislike) and >5 meaning “okay”.

### 2.11. Data Analysis

SPSS statistical software for Windows, version 12, was used to analyze the data. The results were expressed as the mean s.e.m. of three separate studies. To compare differences in means between groups, one-way analysis of variance (ANOVA) was employed, followed by the Tukey test for multiple comparisons. A *p*-value of (0.05) was used to determine whether differences in means between groups were significant.

During storage, samples were statistically analyzed using characteristics such as microbiological counts, chemical oxidation, and sensory features. Before chemometric analyses, all variables were auto-scaled. Principal component analysis (PCA) and heat maps were performed on samples at various storage days using SPSS 26. Dendrograms were created using the XLSTAT software for Windows to provide two-dimensional projections of the various sample groups.

## 3. Results

### 3.1. Antibacterial Activity

The antibacterial activity might be assessed by measuring the inhibition zone (IZ) and determining the MIC and MBC. Table 1 show that TA had varying amounts of antibacterial activity against all of the strains tested. The inhibitory zones measured between 15–20 mm. *L. monocytogenes* (20 mm) had the highest inhibitory zone among G^+^ bacteria, followed by *B. cereus* (18 mm) and *S. aureus* (20 mm). *P. aeruginosa* has the highest inhibitory zone among G^−^ bacteria (18 mm). The inhibitory zone for gentamicin (10 g/well), which was utilized as a positive control for bacteria, was 12 to 25 mm wide. In the negative, there was no inhibitory effect against the germs tested. Thiamine has a higher antibacterial activity against G^+^ bacteria than G^−^ bacteria, with MIC values of 62.5–700 g/mL and 225–750 g/mL, respectively, against the eight G^+^ bacteria tested. Thiamine also inhibited *B. cereus*, *L. monocytogenes*, and *S. aureus* with MICs of 125, 750, and 1000 g/mL, respectively. TA (IZ = 20 mm) was found to considerably suppress the growth of *L. monocytogenes* in the current study.

### 3.2. Effect of Thiamine on Cell Viability

MTT assay was used to test the effect of TA on cell viability. RAW 264 Incubation of 7 cells with or without LPS for 18 h was done. Figure 1 shows that TA (10, 20 and 50 µg/mL) had no effect on cell viability.

### 3.3. Effects of TA on LPS-Induced NO Production

The level of NO in RAW264.7 was measured after treatment with TA. Figure 2 shows that NO production suppression was related to TA concentration, with TA (50 g/L) significantly (P0.01) lowering NO levels in RAW264.7 cells compared to LPS-stimulated RAW264.7 cells.

### 3.4. Effects of TA on LPS-Induced PGE2 Production

PGE2 production was reduced by TA (50 µg/L) in a concentration-dependent manner. TA considerably reduced PGE2 generation when compared to control (LPS-stimulated RAW264.7 cells) (Figure 3C).

### 3.5. Effects of TA on LPS-Induced TNF-α, IL-1β, IL-6 and IL-10

Stimulation of RAW264.7 cells with LPS induced an increased expression of TNF-α, IL-1β, and IL-6. However, treatment of RAW264.7 cells with LPS in the presence of TA produced concentration-dependent decrease in the expression of TNF-α, IL-1β, and IL-6. LPS-stimulated RAW264.7 cells were characterized by IL-10 expression decrease. However, TA treatment restored IL-10 expression (Figure 3A–D).

### 3.6. Effects of TA on NF-𝜅B Activity and Attenuates the Expression of iNOS and COX-2 in LPS-Induced RAW264.7 Cells

Western blot was used to examine NF-KB p65, a key subunit of NF-kB, in the presence or absence of TA. In the absence of TA, NF-KB p65 expression increased significantly, as seen in (Figure 4). In cells treated with TA, however, p65 was suppressed in a concentration-dependent manner.

The effect of TA on the expression of iNOS and COX-2 in LPS-induced RAW264.7 cells was investigated. Before adding LPS (0.5 µg/mL), cells were treated with TA (10, 20, or 50 µg/mL) for 1 h. LPS stimulation for 24 and 7 h increased iNOS and COX-2 expression significantly. However, in TA pre-treated cells, this high expression was considerably reduced in a concentration-dependent manner (Figure 4).

### 3.7. Bio-Preservative Effect of Thiamine (TA) on Raw Minced Beef Meat at 4 °C

#### 3.7.1. Physicochemical Analyses

The effect of TA on the pH of raw minced beef meat over a 14-day storage period at 4 °C shown in (Table 2). TA-treated samples showed a substantial (*p* < 0.05) increase in pH when compared to control (pH 6.90 ± 0.14), with the lowest for TA (0.5%) being 6.10 ± 0.10.

#### 3.7.2. TBARS Value

Table 2 shows that TA treatment had a substantial (*p* < 0.05) effect on TBARS readings at the end of storage. In 1.75 ± 0.06 and 1.50 ± 0.05 mg of malondialdehyde/kg of sample, respectively, they were higher in control samples (C) than in TA-treated samples.

#### 3.7.3. Microbiological Evaluation

The addition of TA caused a significant (P0.05) drop in TVC counts rate (Table 3). At day 10, the control TVC had grown to 7.30 log CFU/g. TVC in T1 (0.1 percent) and T2 (0.5%) were 6.66 and 6.25 log CFU/g, respectively, on the fourteenth day of storage. The results demonstrated a significant decrease in TVC (*p* < 0.05) that was related to the increase in TA concentration.

T1 and T2 TVC did not surpass 6.7 log CFU/g of beef till the end of storage (AFNOR V01-003, 2004), but control sample shelf life was just seven days. With greater levels in control samples, there was a significant increase in PTC (*p* < 0.05) (Table 3). The inclusion of TA effectively reduced PTC. T1, and T2, respectively, slowed PTC to 4.45 and 4.10 log CFU/g at the end of storage, extending shelf life up to 14 days during refrigerated storage. The antibacterial potential of TA was also tested on the number of Enterobacteriaceae. Counts were less than 1.0 log U/g on day 0 and remained below the detection limit (10^2^ CFU/g) for the TA sample (AFNOR 2004) throughout the storage period. On day 14, lower counts characterized treated samples with 0.5 % of TA (T2) showed (*p* < 0.05) compared with the other ones.

### 3.8. Sensory Evaluation

During storage, there was a substantial decline in all sensory qualities (*p* < 0.05). Until 14 days, the treated samples were more stable (limit of rejection was of 5).

Table 4 shows that BHT, T1, and T2 treated minced beef meat were acceptable until 14 days (*p* < 0.05), whereas untreated samples (C) became undesirable after 7 days (*p* < 0.05). Numerous studies [28,29,30] have found a minor reduction in sensory characteristics.

### 3.9. Effect of Thiamine on Viable Counts of Listeria Monocytogenes (ATCC 19117)

The effect of TA on *Listeria monocytogenes* (ATCC 19117), one of the most common causes of food deterioration, was examined. Different concentrations of TA were used to track the kinetics (1 MIC, 2 MIC and 3 MIC). The number of *L. monocytogenes* in all samples at day 0 was similar, as shown in (Figure 5). TA-treated samples showed a substantial (*p* < 0.05) drop in *L. monocytogenes* count compared to untreated samples, especially at concentrations 2 × MIC and 3 × MIC of TA, where the *L. monocytogenes* count was reduced by 2.5 log cycles over a (4–6) day storage period at 4 °C. *L. monocytogenes* growth was similarly impacted by storage duration (*p* < 0.05) (Figure 5). Until the end of the trial, the number of *L. monocytogenes* remained low (10 days). *Listeria monocytogenes* may be delayed (*p* < 0.05) by TA addition at 2 and 3 MIC during raw minced meat beef storage at 4 °C.

### 3.10. Chemometric Analysis

#### 3.10.1. PCA

PCA analysis was used to confirm the clustering of the samples. PCA accounted for 97.07% of the variation in the original data (F1: 88.67%, F2: 8.40%) (Figure 6A). It discovered a link between physicochemical (TBARS and pH) and microbiological (PTC, TVC, and Enterobacteriaceae) characteristics, as well as a link between shorter storage duration (0 and 3 days) and higher sensory scores in T1, T2, and BHT samples (Figure 6B). The advancement of microbial contamination and the accumulation of lipid oxidation products in the meat after storage at 4 °C is represented by the arrangement of samples towards the right side of PCA. On days 10 and 14, a strong link between meat deterioration and the control sample was discovered. However, high odor, color, and acceptability scores were directly associated with TA (T1 and T2) treatments even when storage time was increased. As a result, adding TA to minced beef prevents microbiological and physicochemical deterioration while also preserving sensory qualities.

#### 3.10.2. Heat Map

In each day of storage (3, 7, 10, 14 days), heat maps were utilized to summarize the quantitative data of the samples in terms of lipid/protein oxidation, microbial growth, and sensory characteristics (Figure 7). Each parameter has a different color assigned to it, ranging from red for low concentrations to green for high concentrations. On the tenth day of storage, the difference between the different samples was discovered. As a result, the samples are divided into two clusters: clusters (T1 and T2) and cluster T3; (T1, T2 and BHT). Those two clusters contrast with the third, which is represented by the control and is characterized by the most severe microbial contamination, primarily with PTC and APC. These microbiological factors are, in turn, linked to a decrease in beef sensory quality. This shows that the treatments have a protective impact against microbiological, physicochemical, and, as a result, sensory changes in meat over time.

## 4. Discussion

Findings from the present study show that TA demonstrates anti-inflammatory and anti-bacterial activity against a panel of food spoilage bacteria, suppresses lipid oxidation, and improves shelf-life of raw minced beef meat by enhancing its microbiological, physico-chemical, and sensory characteristics. Several free radicals and inflammatory cytokines play a role in inflammation. Excess or continuous production of these inflammatory mediators can cause a variety of illnesses [31]. Activated macrophages involved in immunological reactions secrete inflammatory cytokines during inflammation [32]. Furthermore, numerous studies have shown that LPS can cause the generation of inflammatory free radicals. TA has been utilized, with encouraging results, to protect nerve cells from a range of poisons, including paraquat and -amyloid [33]. Thiamine has also been discovered to help cells recover from the effects of heavy exercise [34]. In this investigation, TA was found to have an anti-inflammatory impact in LPS-stimulated RAW 264.7 cells. We discovered that TA greatly reduced the production of pro-inflammatory cytokines generated by LPS. Furthermore, it reduced NF-kB activity in vitro. These findings suggest that TA could be a promising new treatment option for inflammatory disorders.

Nuclear factor-kB (NF-kB) is a nuclear protein that regulates the production of inflammation-related factors. The transcription of inflammatory cytokines and enzymes is regulated by activated NF-B [35]. The transcriptional capacity of NF-B is mostly regulated by NF-B p65, a signal for NF-B activation. NF-Bp65 can boost pro-inflammatory cytokines when activated and phosphorylated [35]. As a result, the NF-B signalling pathway could be used to treat inflammatory illnesses. According to the current findings, TA inhibited p65 nuclear translocation in a concentration-dependent manner (Figure 4). Inflammatory illnesses are marked by the pro-inflammatory cytokines NO and PGE2. PGE2 is produced by the enzyme cyclooxygenase (COX). As a result, decreasing NO and PGE2 is a viable method for reducing inflammation [36]. Via immune-histochemical labeling, we discovered that TA dramatically suppressed NO and PGE2 production (Figure 2), as well as downregulating iNOS and COX-2 (Figure 4). Thus, the anti-inflammatory activity of TA could be attributed to limiting the expression of iNOS and COX-2, which inhibits NO and PGE2.

Pro-inflammatory mediators such as TNF-, IL-6, and IL-1β play essential roles in mediating and regulating inflammation in vitro and in vivo. TNF- regulates certain adhesion molecules, which can enhance inflammatory activity.

IL-1β was one of the first pro-inflammatory cytokines to be discovered [37]. The physiopathology of rheumatoid arthritis is heavily influenced by the overexpression of both IL-6 and IL-1 β [38]. TNF-, IL-1 β, and IL-6 were all significantly raised by LPS in this study, whereas TA reduced them (Figure 3A–C). IL-10 is a cytokine that has anti-inflammatory and immunosuppressive properties. It has a well-known inhibitory effect on the generation of inflammatory cytokines [39]. The anti-inflammatory cytokine IL-10 was shown to be considerably elevated by TA in the current investigation. As a result, the cytokine modulation discovered in this study could be one of the mechanisms underpinning TA’s anti-inflammatory effects.

Recent studies have sought to find a safe and effective replacement to antibiotics as food preservatives [40]. Globally, the emergence of resistant bacteria strains has been a key contributor to treatment failures [41]. Herbs have frequently been suggested as effective agents [42]. Exploration of natural compounds with antibacterial activity and no adverse effects has become vital. The antibacterial activity of TA was tested against G^+^ and G^−^ bacteria in this study. TA demonstrated antibacterial action in varying degrees against all microorganisms tested. The bacteria that were investigated were the most common causative organisms for human infections and food spoilage. These results are in agreement with the study of EJ et al. [43]. *B. cereus* and *S. aureus* are enterotoxin-producing bacteria that are resistant to many antibiotics [44]. G^+^ bacteria were more sensitive to TA in this investigation than G^−^ bacteria, which is consistent with a previous study [45]. The structure and makeup of the bacterial cell membrane have been linked to differences in sensitivity to antimicrobial drugs in G^+^ and G^−^ bacteria [46]. As previously stated [47], it is probable that TA destroyed the cell membrane integrity of all sensitive bacteria.

Plants biosynthesize various phytocompounds to fight their natural enemies, mostly microbial invasion and herbivory. These plant-derived phytocompounds demonstrate antimicrobial properties and as a result form part of pre- or post-infection defensive mechanisms of most plants [47]. Interestingly, the microbes that invade plants are the same kind of microbes that contaminate foods, causing food spoilage and also food-borne diseases. In view of this, plant-derived phytocompounds have become useful agents for preventing food spoilage and food-borne diseases. Many microbes are implicated in food spoilage, including G+ and G- bacteria [47]. *Listeria monocytogenes* is one of the bacteria causally associated with food spoilage. It is widely distributed in the environment and has the ability to contaminate a wide range of foods and food sources, including meat, dairy products, and vegetables.

In the present study, incorporation of TA in raw minced beef meat over a 14-day storage period suppressed *L. monocytogenes* growth, indicating that TA has bactericidal activity against *L. monocytogenes*. Additionally, after 14 days, the PTC counts of T1 and T2 were above 4.45 and 4.10 log10 CFU/g, respectively. These values were in the limits set by ISO 4833 (2003) and ICMSF (1986) for PTC in minced meat beef. Compared to TA-treated samples, PTC for control samples of 7.40 log10 CFU/g was quite high and at a value considered as inappropriate for consumption according to ISO 4833. In agreement with the present observations, it was shown that extracts from leaves of *Moringa oleiferi* incorporated in beef patties and herbal chicken sausage at 4 °C for five weeks resulted in a significant reduction in their APC. This finding is consistent with a previous study, which showed that bacteriocin-preserved beef products enhanced their shelf life during cold storage [48]. Similarly, phenolic compounds derived from *Ephedra alata* demonstrated efficacy against foodborne diseases [49]. Also, extracts from *Adansonia digitata* decreased APC during preservation of beef patties [29]. Peel extracts of pomegranate decreased APC (6.23 log CFU/g) over a 21-day storage period of beef patties [50].

Incorporation of TA in fresh minced meat beef improved physico-chemical characteristics and sensory quality of minced beef and also prevented lipid oxidation during storage. The pH of samples C, BHT, T1, and T2 decreased from initial values to 6.90, 6.50, 6.35, and 6.10, respectively. The ability of TA to decrease pH indicates that it prevented the growth and multiplication of most alkaliphiles that cause food spoilage. Also, the antibacterial activity of TA may be attributed to proteolytic enzymes released by the minced beef as well as released peptides.

Lipid oxidation occurs when oxygen reacts with unsaturated lipids, resulting in conjugated dienes, hydroperoxides, and aldehydes. This causes rancidity and the formation of off-flavors in foods [51]. TA-treated samples had lower TBARS levels compared to control samples, indicating that TA prevented lipid oxidation. This observation that TA decreased levels of TBARS perhaps suggests that the TBARS index of approximately 2.0 mg/kg could be used as a criterion for oxidized beef acceptability [52]. Antioxidant activity has been shown in proteins from a variety of sources, which may help to prevent the production of TBARS in beef products [53,54,55,56,57]. Put together, TA treatment of samples (raw minced beef) inhibited microbial growth, decreased lipid oxidation, improved sensory characteristics, and extended the shelf life of raw minced beef during storage at 4 °C.

## 5. Conclusions

TA demonstrated anti-inflammatory in LPS-induced macrophages; anti-bacterial activity against a panel of food spoilage bacteria; anti-lipid oxidation in raw minced beef meat by significant decrease in TBARS; and improved shelf-life of raw minced beef meat by enhancing its microbiological, physico-chemical, and sensory characteristics. Specifically, TA decreased the levels of pro-inflammatory cytokines and associated mediators, including IL-1, IL-6, TNF-, NO, iNOS, and COX-2. Also, TA reduced lipid oxidation and decreased total viable colonies, psychrophilic bacteria, and Enterobacteriaceae in raw minced beef meat. The observed findings with regards to TA provide a strong rationale for consideration of TA as a potential anti-inflammatory agent and also as natural preservative for the meat industry.

## Figures and Tables

**Figure 1 animals-12-01646-f001:**
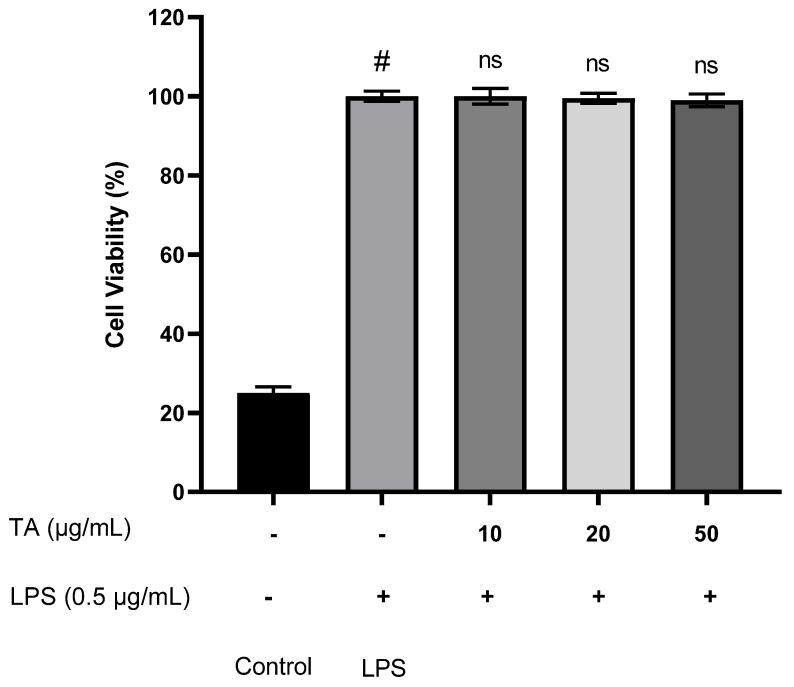
Cytotoxic effect of TA on RAW 264.7 cells. Cells were incubated with increasing concentrations of TA (0, 10, 20 and 50 μg/mL) and incubated overnight at 37 °C in a humidified atmosphere containing 5% CO_2_. Each bar is the mean ± s.e.m, n = 3. ^#^ *p* ≤ 0.05 (control versus LPS treatment); ^ns^ *p* ≥ 0.05 (TA treatments versus LPS treatment). LPS—Lipopolysaccharide, TA—Thiamine.

**Figure 2 animals-12-01646-f002:**
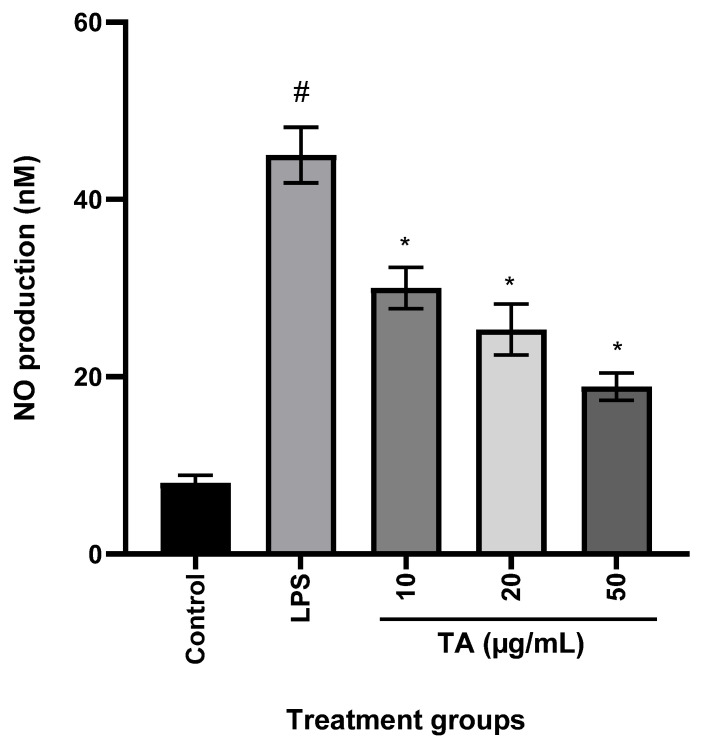
Effect of TA on nitric oxide (NO) production in LPS-stimulated RAW264.7 cells. Each bar is the mean ± s.e.m, n = 3. Cells were incubated in the presence of TA or in the presence of a combination of LPS (0.5 µg/mL) and TA (10, 20, and 50 µg/mL) for 18 h. Data are presented as percentages and LPS control (without TA) was fixed at 100%. ^#^ *p* ≤ 0.05 (control versus LPS treatment); * *p* ≤ 0.05 (TA treatments versus LPS treatment). One-way ANOVA followed by Dunnett’s post-hoc test. NO (IC_50_ = 0.067 µg/mL). LPS—lipopolysaccharide.

**Figure 3 animals-12-01646-f003:**
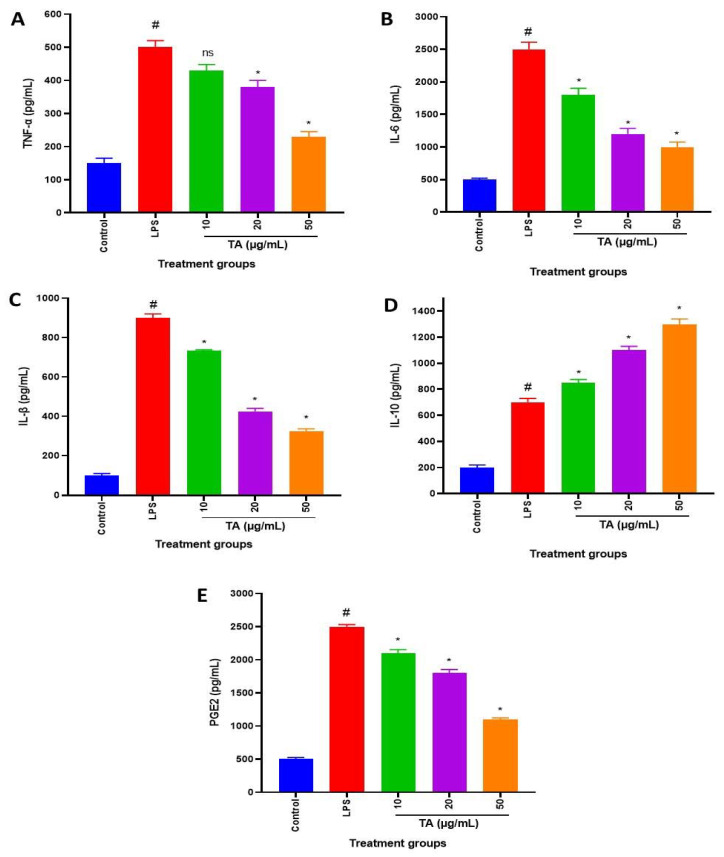
Effects of TA on (**A**) TNF-α, (**B**) IL-6, (**C**) IL-β, (**D**) IL-10 and (**E**) PGE2 in LPS-stimulated RAW264.7 cells. The cells were pretreated with increasing concentrations of TA (10, 20, and 50 µg/mL) for 1 h and then exposed to LPS (0.5 µg/mL) for 18 h. The levels of TNF-α, IL-6, IL-β, IL-10 and PGE2 in cell extracts were determined by ELISA. Each bar is the mean ± s.e.m, n = 3. ^#^ *p* ≤ 0.05 (control versus LPS treatment); * *p* ≤ 0.05, ^ns^ *p* ≥ 0.05 (TA treatments versus LPS treatment). TNF-α (IC_50_ = 6.39 µg/mL), IL-6 (IC_50_ = 5.58 µg/mL), IL-1β (IC_50_ = 5.18 µg/mL), and PGE2 (IC_50_ = 6.55 µg/mL).

**Figure 4 animals-12-01646-f004:**
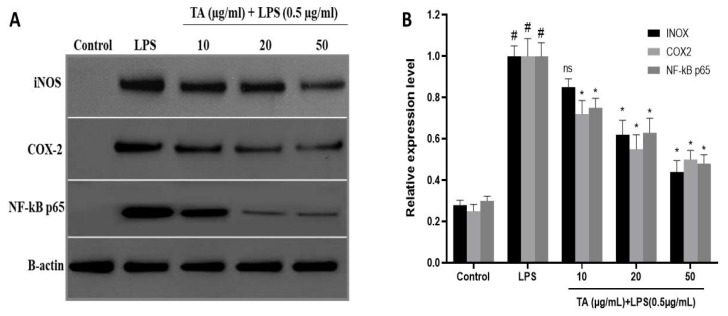
Effect of TA on protein expression of (**A**) iNOS, COX-2 and NF-κB (p65) in LPS-stimulated RAW 264.7 cells, and (**B**) a semi-quantitative measurement of iNOS, COX-2, and NF-κB (p65). β-actin was used as an internal control. Cells were pre-treated with increasing concentrations of TA (0, 10, 20, and 50 μg/mL) for 2 h and then stimulated with LPS (0.5 μg/mL) or without LPS for 30 min at 37 °C in a humidified atmosphere containing 5% CO_2_. (**A**) is a representative of three independent experiments. (**B**) Each bar is the mean ± s.e.m, n = 3. ^#^ *p* ≤ 0.05 (control versus LPS treatment); * *p* ≤ 0.05, ^ns^ *p* ≥ 0.05 (TA treatments versus LPS treatment). LPS—Lipopolysaccharide; TA—Thiamine.

**Figure 5 animals-12-01646-f005:**
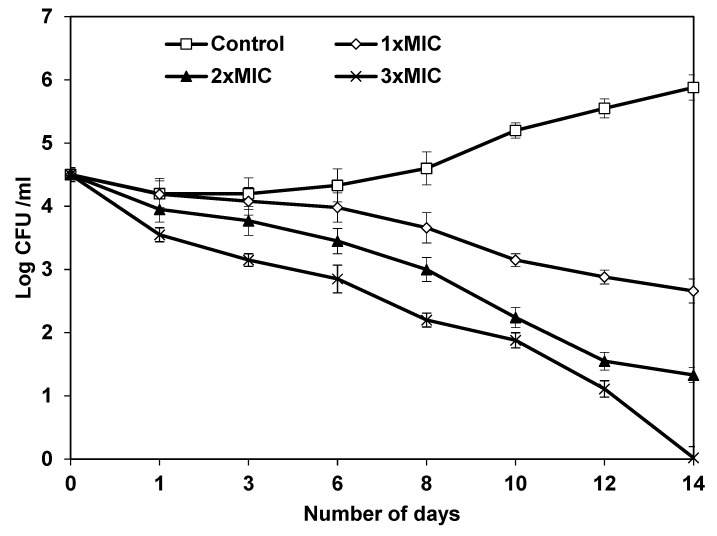
Time-related survival of *Listeria monocytogenes* at 4 °C following treatment with increasing concentrations of TA. Each plotted point is the mean ± s.e.m, n = 3. Bacteria were supplemented in minced beef meat samples at 6 × 10^5^ CFU/g of meat. MIC—Minimum Inhibitory Concentration; TA—Thiamine.

**Figure 6 animals-12-01646-f006:**
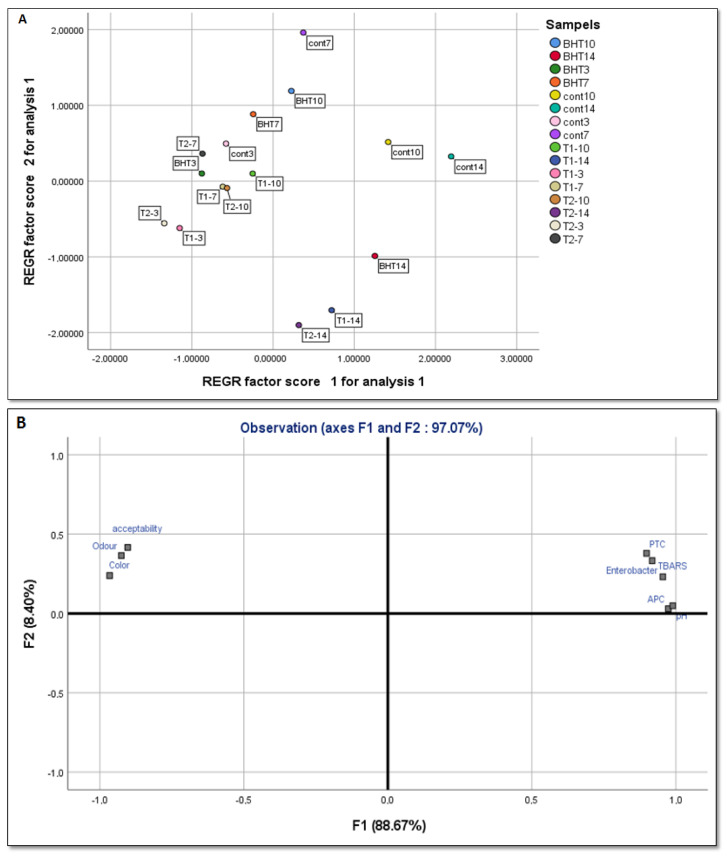
Principal component analysis (PCA) plots of physicochemical parameters, microbial loads, and sensory characteristics of different samples at each storage time: (**A**) variable-loading plot of PCA; (**B**) observation score plot of PCA.

**Figure 7 animals-12-01646-f007:**
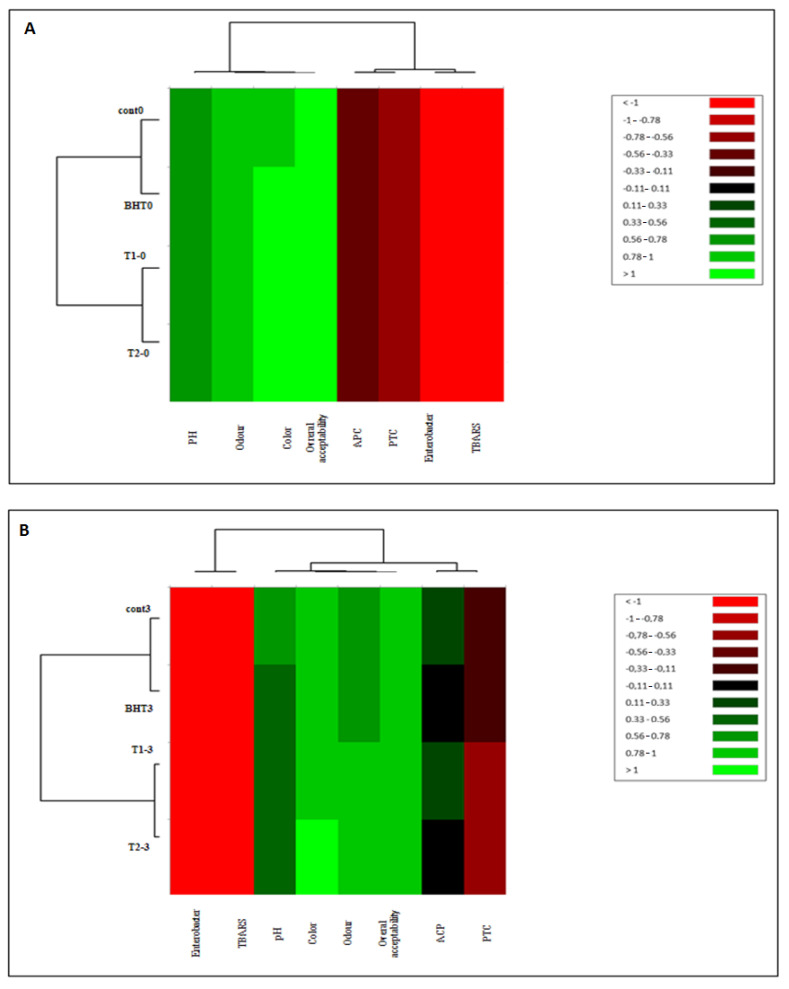
Agglomerative hierarchical cluster analysis (HCA) and heat map of physicochemical characteristics, microbial counts, and sensory attributes of untreated (Control and BHT) and different treated (T1 and T2) samples at each storage time (days): [0 (**A**); 3 (**B**); 7 (**C**); 10 (**D**) and 14 (**E**)] days.

**Table 1 animals-12-01646-t001:** Antibacterial activity of thiamine (TA) against foodborne, spoiling bacteria and determination of the minimum inhibitory concentrations (MICs) and minimum bactericidal concentration (MBC).

Bacterial Strains	Zones of Inhibition (mm) ^a^			
	TA ^b^	Gentamicin ^c^	MIC (µg/mL)	MBC (µg/mL)
Gram positive				
*Bacillus subtilis* ATCC 6633 *Bacillus cereus* ATCC 14579	15 ± 0.4 18 ± 0.7	20 ± 0.2 20 ± 0.4	500 ± 1.2 750 ± 0.9	1000 750
*Staphylococcus aureus* ATCC 25923 *Staphylococcus aureus* ATCC 6536 *Staphylococcus epidermis* ATCC 12228	20 ± 0.9 20 ± 0.6 17 ± 0.5	25 ± 0.8 16 ± 0.6 20 ± 0.5	250 ± 1.3 250 ± 1.3 125 ± 1.5	250 500 250
*Enterococcus faecalis* ATCC 29212 *Micrococcus luteus* ATCC 1880 *Listeria monocytogenes* (food isolate 2132)	18 ± 0.4 15 ± 0.2 20 ± 0.9	12 ± 0.2 20 ± 0.7 15 ± 0.0	62.5 ± 1.3 250 ± 1.6 125 ± 1.1	250 250 500
Gram negative				
*Salmonella enteritidis* (food isolate) *Escherichia coli* ATCC 25922 *Escherichia coli* ATCC 8739	18 ± 1.1 15 ± 0.5 15 ± 0.5	18 ± 0.8 21 ± 1.0 20 ± 1.2	450 ± 1.7 750 ± 1.9 225 ± 1.2	450 750 500
*Pseudomonas aeruginosa* ATCC 9027	18 ± 0.4	18 ± 0.7	225 ± 1.0	750
*Klebsiella pneumoniae* ATCC 10031	17 ± 0.4	12 ± 0.5	750 ± 0.9	1000

Values are given as mean ± S.D of triplicate experiments. ^a^ Diameter of zones of inhibition including diameter of disc (6 mm). ^b^ TA—Thiamine (50 µL/mL).^c^ Gentamicin (10 μg/mL).

**Table 2 animals-12-01646-t002:** Effect of TA on pH and TBARS (mg of malonaldehyde equivalents per kg of sample (mg/kg) values of raw minced meat beef during storage at 4 °C.

			Days of Storage at 4 °C		
	0	3	7	10	14
pH					
Control	5.60 ± 0.16 ^a^	5.80 ± 0.12 ^b^	6.20 ± 0.11 ^c^	6.50 ± 0.14 ^d^	6.90 ± 0.14 ^d^
BHT	5.60 ± 0.11 ^a^	5.60 ± 0.12 ^a^	6.00 ± 0.12 ^b^	6.20 ± 0.14 ^b^	6.50 ± 0.15 ^b^
T_1_	5.60 ± 0.10 ^a^	5.50 ± 0.15 ^a^	5.85 ± 0.11 ^a^	6.00 ± 0.15 ^a^	6.35 ± 0.12 ^a^
T_2_	5.60 ± 0.09 ^a^	5.45 ± 0.09 ^a^	5.70 ± 0.21 ^a^	5.85 ± 0.11 ^a^	6.10 ± 0.10 ^a^
TBARS					
Control	0.20 ± 0.00 ^a^	1.30 ± 0.08 ^c^	2.00 ± 0.09 ^c^	2.50 ± 0.09 ^e^	2.90 ± 0.15 ^c^
BHT	0.20 ± 0.00 ^a^	0.90 ± 0.03 ^ab^	1.50 ± 0.06 ^b^	1.65 ± 0.07 ^c^	1.85 ± 0.09 ^b^
T_1_	0.20 ± 0.00 ^a^	0.52 ± 0.07 ^a^	1.20 ± 0.04 ^b^	1.45 ± 0.08 ^b^	1.75 ± 0.06 ^b^
T_2_	0.20 ± 0.00 ^a^	0.32 ± 0.04 ^a^	0.95 ± 0.03 ^a^	1.15 ± 0.03 ^a^	1.50 ± 0.05 ^a^

Values are given as mean ± standard deviation of triplicate experiments. ^a–e^: Averages with different letters in the same column, for each parameter, are different (*p* < 0.05). (C) Control, (BHT) butylated hydroxytoluene, (T_1_) treatment with 0.1% TA and (T_2_) treatment with 0.5% TA.

**Table 3 animals-12-01646-t003:** Effect of TA on the microbial load of aerobic plate count (APC), psychotropic count (PTC) and *Enterobacteriaceae* count (log10 CFU/g) of raw minced meat beef during storage at 4 °C.

	Days of Storage at 4 °C				
	0	3	7	10	14
Total viable count					
Control	2.40 ± 0.10 ^a^	4.80 ± 0.30 ^d^	6.66 ± 0.22 ^e^	7.30 ± 0.31 ^f^	8.50 ± 0.33 ^e^
BHT	2.40 ± 0.10 ^a^	4.60 ± 0.15 ^c^	5.70 ± 0.19 ^c^	6.25 ± 0.20 ^c^	7.60 ± 0.20 ^b^
T_1_	2.40 ± 0.15 ^a^	4.50 ± 0.20 ^b^	5.83 ± 0.22 ^b^	5.90 ± 0.15 ^b^	6.66 ± 0.10 ^b^
T_2_	2.40 ± 0.10 ^a^	4.30 ± 0.16 ^a^	5.20 ± 0.10 ^a^	5.45 ± 0.20 ^a^	6.25 ± 0.25 ^a^
Psychotropic count					
Control	2.10 ± 0.03 ^a^	4.20 ± 0.15 ^d^	5.80 ± 0.20 ^e^	6.50 ± 0.22 ^e^	7.40 ± 0.30 ^f^
BHT	2.10 ± 0.03 ^a^	4.10 ± 0.13 ^b^	5.10 ± 0.10 ^c^	5.95 ± 0.15 ^c^	6.15 ± 0.15 ^c^
T_1_	2.10 ± 0.04 ^a^	2.90 ± 0.10 ^a^	3.20 ± 0.17 ^b^	4.20 ± 0.18 ^b^	4.45 ± 0.11 ^b^
T_2_	2.10 ± 0.01 ^a^	2.80 ± 0.17 ^a^	3.09 ± 0.07 ^a^	3.85 ± 0.15 ^a^	4.10 ± 0.20 ^a^
Enterobacteriaceae count					
Control	<1	1.85 ± 0.09 ^c^	2.85 ± 0.14 ^c^	3.30 ± 0.10 ^d^	3.55 ± 0.05 ^c^
BHT	<1	1.50 ± 0.10 ^a^	1.85 ± 0.10 ^a^	2.30 ± 0.14 ^b^	2.55 ± 0.09 ^b^
T_1_	<1	1.40 ± 0.10 ^a^	1.60 ± 0.08 ^a^	1.75 ± 0.10 ^b^	2.09 ± 0.07 ^b^
T_2_	<1	1.30 ± 0.08 ^a^	1.45 ± 0.07 ^a^	1.54 ± 0.10 ^a^	1.69 ± 0.09 ^a^

Values are given as mean ± Standard deviation of triplicate experiments ^a–f^: averages with different letters in the same column, for each parameter, are different (*p* < 0.05). (C) control, (BHT) butylated hydroxytoluene, (T_1_) treatment with 0.1% TA and (T_2_) treatment with 0.5% TA.

**Table 4 animals-12-01646-t004:** Effect of TA on color, odour and overall acceptability of raw minced meat beef stored at 4 °C.

	Days of Storage at 4 °C				
	0	3	7	10	14
Color					
Control	6.30 ± 0.25 ^a^	6.10 ± 0.15 ^a^	5.50 ± 0.11 ^a^	3.50 ± 0.10 ^a^	2.50 ± 0.08 ^a^
BHT	6.50 ± 0.23 ^b^	6.50 ± 0.14 ^b^	6.10 ± 0.11 ^b^	5.45 ± 0.20 ^c^	3.70 ± 0.09 ^b^
T_1_	6.80 ± 0.16 ^c^	6.55 ± 0.20 ^b^	6.30 ± 0.20 ^b^	5.85 ± 0.20 ^c^	3.85 ± 0.04 ^b^
T_2_	6.80 ± 0.17 ^c^	6.85 ± 0.20 ^c^	6.65 ± 0.20 ^c^	6.10 ± 0.17 ^d^	4.20 ± 0.05 ^c^
Odour					
Control	6.10 ± 0.18 ^a^	6.00 ± 0.13 ^a^	5.75 ± 0.11 ^a^	3.60 ± 0.10 ^a^	2.20 ± 0.01 ^a^
BHT	6.20 ± 0.14 ^b^	6.00 ± 0.17 ^a^	5.80 ± 0.21 ^c^	5.20 ± 0.19 ^b^	2.65 ± 0.06 ^c^
T_1_	6.30 ± 0.11 ^b^	6.15 ± 0.21 ^b^	6.00 ± 0.19 ^c^	5.30 ± 0.22 ^b^	3.25 ± 0.04 ^c^
T_2_	6.60 ± 0.19 ^c^	6.50 ± 0.22 ^c^	6.50 ± 0.18 ^c^	5.65 ± 0.23 ^b^	3.85 ± 0.02 ^c^
Overall acceptability					
Control	6.50 ± 0.22 ^a^	6.20 ± 0.20 ^a^	5.80 ± 0.19 ^a^	3.30 ± 0.14 ^a^	2.65 ± 0.04 ^a^
BHT	6.50 ± 0.19 ^b^	6.30 ± 0.18 ^c^	6.00 ± 0.17 ^c^	5.75 ± 0.23 ^c^	2.85 ± 0.09 ^b^
T_1_	6.60 ± 0.22 ^b^	6.30 ± 0.19 ^c^	6.15 ± 0.11 ^d^	5.85 ± 0.1 ^d^	3.25 ± 0.08 ^c^
T_2_	6.70 ± 0.22 ^c^	6.50 ± 0.16 ^d^	6.50 ± 0.21 ^d^	6.10 ± 0.20 ^d^	3.50 ± 0.08 ^c^

Values are given as mean ± standard deviation of triplicate. ^a–d^: averages with different letters in the same column, for each parameter, are different (*p* < 0.05). (C) Control, (BHT) butylated hydroxytoluene, (T_1_) treatment with 0.1% TA and (T_2_) treatment with 0.5% TA.

## Data Availability

Not applicable.

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
