# Peer review of "Thiamine Demonstrates Bio-Preservative and Anti-Microbial Effects in Minced Beef Meat Storage and Lipopolysaccharide (LPS)-Stimulated RAW 264.7 Macrophages"

_animals, 2022, doi:10.3390/ani12131646_

Round 1

Reviewer 1 Report

Dear Madam / Sir

This manuscript is about the antioxidant and antimicrobial effects of thiamine (TA). The antioxidant properties of TA were screened in a macrophage cell line, whereas the antimicrobial activities were screened both in vitro and in minced meat. The manuscript is generally well written. Still it is quite difficult to understand the link between 1) the research in macrophages 2) the in vitro antimicrobial activity on specific strains 3) the effect on minced meat storage. In addition, beef is considered a rich source of TA. Therefore supplementation with TA (a quite heat labile substance) is of questionable benefit when administered through meat which is generally heat treated before consumption, if the beneficial antioxidant and anti-inflammatory properties are to be sought for. The antimicrobial experiments are valid and well understood under the prism of the prolonged minced meat storage, even though the microbial counts were quite high after 14 days, making minced meat possibly very close to spoiled at this time. Even so, the main problem of this manuscript is that it does not provide a link between the in vitro experiments (especially the ones concerning the antioxidant activities on macrophages) with the possibility of TA use in meat preservation.

Specific comments.

Title. The title is not appropriate and generally not informative. TA is not mentioned in it.

P2 L48-9. Food spoilage is not a cause of foodborne disease. Food contamination with pathogenic microrganisms is the cause.

P2 L67. iNOs have not been mentioned elsewhere.

P2L75-80. In the Discussion part, several articles are discussed, possibly because of their rich TA concentration. In order for the reader to follow the authors thoughts, perhaps information concerning TA contents on plants and generally food should be made.

P2 L91. Materials and Methods in general. The reagents companies are missing in most cases. Please supply details.

P3 L112. DMSO is supposed to have some mild antimicrobial activity that could possibly influence the results. Can the authors comment on this?

P3 L118. TA is probably out of place in the start of the sentence.

P3 L126-130. The description of the tetrazolium method does not seem correct. Perhaps there is some details missing. Please rephrase or simplify.

P4 L183-4. 0.1% peptone water is probably not isotonic if only 0.1% peptone was used. Please comment.

P4 L186. Enterobacteriaceae family can be simplified to Enterobacteriaceae.

P4 L190. L. monocytogenes in italics.

P5 L220-221. This sentence belongs to materials and methods.

P5 L232-234. A left over from the description of this section by MDPI. Please omit.

P10 line 328. It is not TVC growth rate that the authors have screened for, but TVC counts.

P15 L424. Discussion in general. Although the results are quite rich, the discussion part in quite limited and fails to actually discuss the findings against other researchers and generally define the frame in which this research fills the possible knowledge gaps.

P15 L450-2. This is most suitable to introduction.

P15 L464. Replace bacterium by bacterial

P15 L473. Please select between G- or Gram-.

P16 L478-519. In this rather large paragraph, several plants are mentioned, such as Ephedra alata, Andasognia digitata, and pomegranate are reported, possibly due to the thiamine content or some related substance. The average reader though needs some relevant information in order to comprehend the comparison of results.

P16 L505-7. This is a possible explanation of the observed antimicrobial activity of TA. Nevertheless, it is not backed up by similar observations by other researchers. Specifically, please comment on the possible induction by TA of proteolysis by beef meat strains (the type of meat flora implicated could be of use) and active substances production.

Author Response

Reviewer 1:

Dear Madam / Sir

This manuscript is about the antioxidant and antimicrobial effects of thiamine (TA). The antioxidant properties of TA were screened in a macrophage cell line, whereas the antimicrobial activities were screened both in vitro and in minced meat. The manuscript is generally well written. Still it is quite difficult to understand the link between 1) the research in macrophages 2) the in vitro antimicrobial activity on specific strains 3) the effect on minced meat storage. In addition, beef is considered a rich source of TA. Therefore supplementation with TA (a quite heat labile substance) is of questionable benefit when administered through meat which is generally heat treated before consumption, if the beneficial antioxidant and anti-inflammatory properties are to be sought for. The antimicrobial experiments are valid and well understood under the prism of the prolonged minced meat storage, even though the microbial counts were quite high after 14 days, making minced meat possibly very close to spoiled at this time. Even so, the main problem of this manuscript is that it does not provide a link between the in vitro experiments (especially the ones concerning the antioxidant activities on macrophages) with the possibility of TA use in meat preservation.

 Specific comments.

- The title is not appropriate and generally not informative. TA is not mentioned in it.

  • We have corrected The title

- P2 L48-9. Food spoilage is not a cause of foodborne disease. Food contamination with pathogenic microrganisms is the cause.

  • We have corrected the sentence.

-P2 L67. iNOs have not been mentioned elsewhere.

  • It is noted in p2 L67 that iNOs is " inducible nitric oxide synthase"

-P2L75-80. In the Discussion part, several articles are discussed, possibly because of their rich TA concentration. In order for the reader to follow the authors thoughts, perhaps information concerning TA contents on plants and generally food should be made.

  • In this study, we focus on the effect of TA as a natural compound. Our goal is not to see the TA content on plants. This is another research work in our laboratory that is in progress.

-P2 L91. Materials and Methods in general. The reagents companies are missing in most cases. Please supply details.

  • We have added the details of reagents companies

-P3 L112. DMSO is supposed to have some mild antimicrobial activity that could possibly influence the results. Can the authors comment on this?

  • We thank the reviewer for this excellent question.
  • All solvents must be having some adverse effect on microbial growth. However, DMSO is generally used to dissolve and assay antibiotic compounds. There is controversy in the literature, where some report not to use more than 2.5% DMSO and few report 100% DMSO can be used to dissolve your extract to testify to its antimicrobial activity. Many researches use 100% DMSO and he may also maintain DMSO as control. In our laboratory, DMSO concentration of 99.8% to 100% is appropriate for antimicrobial assays to yield better results (published protocol).

-P3 L118. TA is probably out of place in the start of the sentence.

  • We have corrected the sentence.

-P3 L126-130. The description of the tetrazolium method does not seem correct. Perhaps there is some details missing. Please rephrase or simplify.

  • We have reworded the sentence.

-P4 L183-4. 0.1% peptone water is probably not isotonic if only 0.1% peptone was used. Please comment.

  • 1% peptone water solution is recommended as a diluent for dilution of sample by different test methods widely used for examination of foodstuffs.
  • We have added others information in the manuscript.

-P4 L186. Enterobacteriaceae family can be simplified to Enterobacteriaceae.

  • It’s done.

-P4 L190. L. monocytogenes in italics.

  • It’s done.

-P5 L220-221. This sentence belongs to materials and methods.

  • It’s done.

-P5 L232-234. A left over from the description of this section by MDPI. Please omit.

  • We have removed this section.

-P10 line 328. It is not TVC growth rate that the authors have screened for, but TVC counts.

  • We have corrected the sentence.

-P15 L424. Discussion in general. Although the results are quite rich, the discussion part in quite limited and fails to actually discuss the findings against other researchers and generally define the frame in which this research fills the possible knowledge gaps.

  • We have improved the discussion.

-P15 L450-2. This is most suitable to introduction.

  • I think this is an introductory sentence to talk about “Nuclear factor-kB” in the discussion section.

-P15 L464. Replace bacterium by bacterial

  • It’s done.

P15 L473. Please select between G- or Gram-.

  • It’s done.

-P16 L478-519. In this rather large paragraph, several plants are mentioned, such as Ephedra alataAndasognia digitata, and pomegranate are reported, possibly due to the thiamine content or some related substance. The average reader though needs some relevant information in order to comprehend the comparison of results.

  • We have improved the paragraph.

-P16 L505-7. This is a possible explanation of the observed antimicrobial activity of TA. Nevertheless, it is not backed up by similar observations by other researchers. Specifically, please comment on the possible induction by TA of proteolysis by beef meat strains (the type of meat flora implicated could be of use) and active substances production.

  • It’s done

Reviewer 2 Report

The Authors of manuscript described the anti-inflammatory effect of thiamine (vitamin B1) in lipopolysaccharide-stimulated RAW264.7 cells and also the preservative properties of TA in minced beef. It is really interesting work, but I have a few comments:

line 190: “L. monocytogenes” should be italic

also line  193 “Listeria” should be italic

in Table 1: should be “Minimal Bactericidal Concentration”, with capital letters

Please explain why in Table 1 the inhibition zones are different as in the chapter 3.1 Antibacterial activity, e.g. L. monocytogenes in the text the IZ is 28mm but in Table 1 20mm, similar situation with B. cereus, S. aureus and P. aeruginosa.

Figure 7: the descriptions of the x and y axes are not readable, please correct.

Author Response

Reviewer 2:

The Authors of manuscript described the anti-inflammatory effect of thiamine (vitamin B1) in lipopolysaccharide-stimulated RAW264.7 cells and also the preservative properties of TA in minced beef. It is really interesting work, but I have a few comments:

-line 190: “L. monocytogenes” should be italic

  • It’s done.

-also line  193 “Listeria” should be italic

  • It’s done.

-in Table 1: should be “Minimal Bactericidal Concentration”, with capital letters

  • It’s done.

-Please explain why in Table 1 the inhibition zones are different as in the chapter 3.1 Antibacterial activity, e.g. L. monocytogenes in the text the IZ is 28mm but in Table 1 20mm, similar situation with B. cereusS. aureus and P. aeruginosa.

  • We have corrected le different value of the IZ in the text.

-Figure 7: the descriptions of the x and y axes are not readable, please correct.

  • We have corrected the descriptions of the x and y axes.

Reviewer 3 Report

The authors have reported the antibacterial, anti-inflammatroy and antioxidative activities or Thiamine or vitamine B1. The antibacterial effect was demonstrated in miced meat (also in case, when the meat was deliberately spiked with Listeria monocytogenes). This finding, although interesting, is not novel. The authors should cite, for example:

Koo EJ, Kwon KH, Oh SW. The antimicrobial effect of thiamine dilauryl sulfate in tofu inoculated with Escherichia coli O157:H7, Salmonella Typhimurium, Listeria monocytogenes and Bacillus cereus. Food Sci Biotechnol. 2017;27(1):283-289.

As an editorial point, the quality of English should be checked, preferably by a native speaker.

In Materials and Methods, the Section 2.6. ( Measurement of NO, PGE2, TNF-?, IL-1?, and IL-6 from Cell Extracts) should be completed, because it is not apparent, what methods were used in the cytokine analysis.

Author Response

Reviewer 3:

-The authors have reported the antibacterial, anti-inflammatroy and antioxidative activities or Thiamine or vitamine B1. The antibacterial effect was demonstrated in miced meat (also in case, when the meat was deliberately spiked with Listeria monocytogenes). This finding, although interesting, is not novel. The authors should cite, for example:

Koo EJ, Kwon KH, Oh SW. The antimicrobial effect of thiamine dilauryl sulfate in tofu inoculated with Escherichia coli O157:H7, Salmonella Typhimurium, Listeria monocytogenes and Bacillus cereusFood Sci Biotechnol. 2017;27(1):283-289.

  • It’s done. The reference has been cites in the manuscript.

-As an editorial point, the quality of English should be checked, preferably by a native speaker.

  • The manuscript has been verified by an English language specialist.

-In Materials and Methods, the Section 2.6. (Measurement of NO, PGE2, TNF-?, IL-1?, and IL-6 from Cell Extracts) should be completed, because it is not apparent, what methods were used in the cytokine analysis.

- We have added references that describe methods were used in the cytokine analysis.

Round 2

Reviewer 1 Report

I would like to thank the authors for their effort.

Reviewer 3 Report

The authors have addressed my main comments. They should, however, correct the reference 43 on line 499. The author is Koo, not "EJ" (these are the intials).